# Nitrogenase resurrection and the evolution of a singular enzymatic mechanism

**Amanda K Garcia[1], Derek F Harris[2], Alex J Rivier[1], Brooke M Carruthers[1], Azul Pinochet-Barros[1], Lance C Seefeldt[2], Betül Kaçar[1]\***

[1]Department of Bacteriology, University of Wisconsin–Madison, Madison, United States; [2]Department of Chemistry and Biochemistry, Utah State University, Logan, United States

**\*For correspondence:**
bkacar@wisc.edu

**Competing interest:** The authors declare that no competing interests exist.

**Abstract** The planetary biosphere is powered by a suite of key metabolic innovations that emerged early in the history of life. However, it is unknown whether life has always followed the same set of strategies for performing these critical tasks. Today, microbes access atmospheric sources of bioessential nitrogen through the activities of just one family of enzymes, nitrogenases. Here, we show that the only dinitrogen reduction mechanism known to date is an ancient feature conserved from nitrogenase ancestors. We designed a paleomolecular engineering approach wherein ancestral nitrogenase genes were phylogenetically reconstructed and inserted into the genome of the diazotrophic bacterial model, *Azotobacter vinelandii*, enabling an integrated assessment of both in vivo functionality and purified nitrogenase biochemistry. Nitrogenase ancestors are active and robust to variable incorporation of one or more ancestral protein subunits. Further, we find that all ancestors exhibit the reversible enzymatic mechanism for dinitrogen reduction, specifically evidenced by hydrogen inhibition, which is also exhibited by extant *A. vinelandii* nitrogenase isozymes. Our results suggest that life may have been constrained in its sampling of protein sequence space to catalyze one of the most energetically challenging biochemical reactions in nature. The experimental framework established here is essential for probing how nitrogenase functionality has been shaped within a dynamic, cellular context to sustain a globally consequential metabolism.

## Editor's evaluation

This manuscript reports valuable findings regarding the evolution of nitrogenases through ancestral sequence reconstruction and resurrection. The results are convincing and support the conclusions of the study, and highlight the historical constraints that have been acting on this enzyme. The findings will be of interest to people interested in enzyme evolution in general and particularly to those interested in the evolution of nitrogenases.

## Introduction

The evolutionary history of life on Earth has generated tremendous ecosystem diversity, the sum of which is orders of magnitude larger than that which exists at present (*Jablonski, 2004*). Life's historical diversity provides a measure of its ability to solve adaptive problems within an integrated planetary system. Accessing these solutions requires a deeper understanding of the selective forces that have shaped the evolution of molecular-scale, metabolic innovations. However, the early histories of many of life's key metabolic pathways and the enzymes that catalyze them remain coarsely resolved.

Important efforts to advance understanding of early metabolic innovations have included phylogenetic inference (*Gold et al., 2017*; *Sánchez-Baracaldo and Cardona, 2020*), systems-level network reconstructions (*Goldford et al., 2017*), and the leveraging of extant or mutant biological models as proxies for their ancient counterparts (*Zerkle et al., 2006*; *Soboh et al., 2010*; *Hurley et al., 2021*). However, methods to directly study these metabolic evolutionary histories across past environmental and cellular transitions remain underexplored. A unified, experimental strategy that integrates historical changes to enzymes, which serve as the primary interface between metabolism and environment, and clarifies their impact within specific cellular and physiochemical contexts is necessary. To address this, phylogenetic reconstructions of enzymes can be directly integrated within laboratory microbial model systems (*Garcia and Kaçar, 2019*; *Kacar et al., 2017b*; *Kędzior et al., 2022*). In this paleomolecular framework, predicted ancestral enzymes can be 'resurrected' within a compatible host organism for functional characterization. These experimental systems can ultimately integrate multiple levels of historical analysis by interrogating critical features of ancient enzymes as well as dynamic interactions between enzymes, their broader metabolic networks, and the external environment.

The study of biological nitrogen fixation offers a promising testbed to thread these investigations of early metabolic evolution. Both phylogenetic and geological evidence (*Garcia et al., 2020*; *Raymond et al., 2004*; *Boyd et al., 2011a*; *Stüeken et al., 2015*; *Parsons et al., 2021*) indicate that the origin of biological nitrogen fixation was a singular and ancient evolutionary event on which the modern biosphere has since been built. The only known nitrogen fixation pathway (compared to, for example, at least seven carbon-fixation pathways [*Garcia et al., 2021a*]) is catalyzed by an early-evolved family of metalloenzymes called nitrogenases that reduce highly inert, atmospheric dinitrogen ($N_2$) to bioavailable ammonia ($NH_3$). The nitrogenase family comprises three isozymes that vary in their metal dependence (i.e. molybdenum, vanadium, and iron) and, in certain cases, all coexist within the same host organism (*Mus et al., 2018*). Many diazotrophs depend on genetic strategies for coordinating the biosynthesis and expression of multiple nitrogenase isozymes and their respective metalloclusters (*Pérez-González et al., 2021*; *Burén et al., 2020*), and, in oxic environments, protecting the oxygen-sensitive metalloclusters from degradation (*Gallon, 1992*). Thus, nitrogenase enzymes are a central component of a broader, co-evolving nitrogen fixation machinery. These features that create significant experimental challenges for nitrogen fixation engineering (*Smanski et al., 2014*; *Bennett et al., 2023*; *Burén and Rubio, 2018*) nevertheless also make this metabolism an ideal candidate for systems-level, paleomolecular study.

How biological nitrogen fixation emerged and evolved under past environmental conditions is still poorly constrained relative to its importance in Earth's planetary and biological history. Because nitrogen has been a limiting nutrient over geological timescales (*Falkowski, 1997*; *Allen et al., 2019*), nitrogenase has long been a key constituent of the expanding Earth's biosphere. The impact of nitrogen limitation is underscored by human reliance on the industrial Haber-Bosch process, an energetically and environmentally costly workaround for nitrogen fertilizer production (*Vicente and Dean, 2017*) designed to supplement a remarkable molecular innovation that biology has tinkered with for more than three billion years. How the structural domains and regulatory network of nitrogenase were recruited (*Boyd et al., 2015*; *Mus et al., 2019*) and under what selective pressures the metal dependence of nitrogenases was shaped (*Garcia et al., 2020*; *Boyd et al., 2011b*) remain open questions. Importantly, it is not known how the enzymatic mechanism for dinitrogen reduction has been tuned by both peptide and metallocluster to achieve one of the most difficult reactions in nature (*Seefeldt et al., 2020*; *Stripp et al., 2022*; *Harris et al., 2019*). At the enzyme level, previous insights into nitrogenase sequence-function relationships have primarily derived from single or dual substitution studies. These have often yielded diminished or abolished nitrogenase activity (*Seefeldt et al., 2020*; *Stripp et al., 2022*), though in certain cases improved reactivity toward alternate, industrially relevant substrates (*Seefeldt et al., 2020*). Despite illuminating key features of extant nitrogenase mechanisms in select model organisms, the combination of detailed functional studies within an explicit evolutionary scheme has not previously been accomplished for the nitrogen fixation system.

Here, we seek guidance from the Earth's evolutionary past to reconstruct the history of the key metabolic enzyme, nitrogenase. We establish an evolutionary systems biology approach for the cellular- and molecular-level characterization of ancestral nitrogenases resurrected within the model diazotrophic bacterium, *A. vinelandii*. We find that variably replacing different protein subunits of the nitrogenase complex with inferred ancestral counterparts enables nitrogen fixation in *A. vinelandii*.

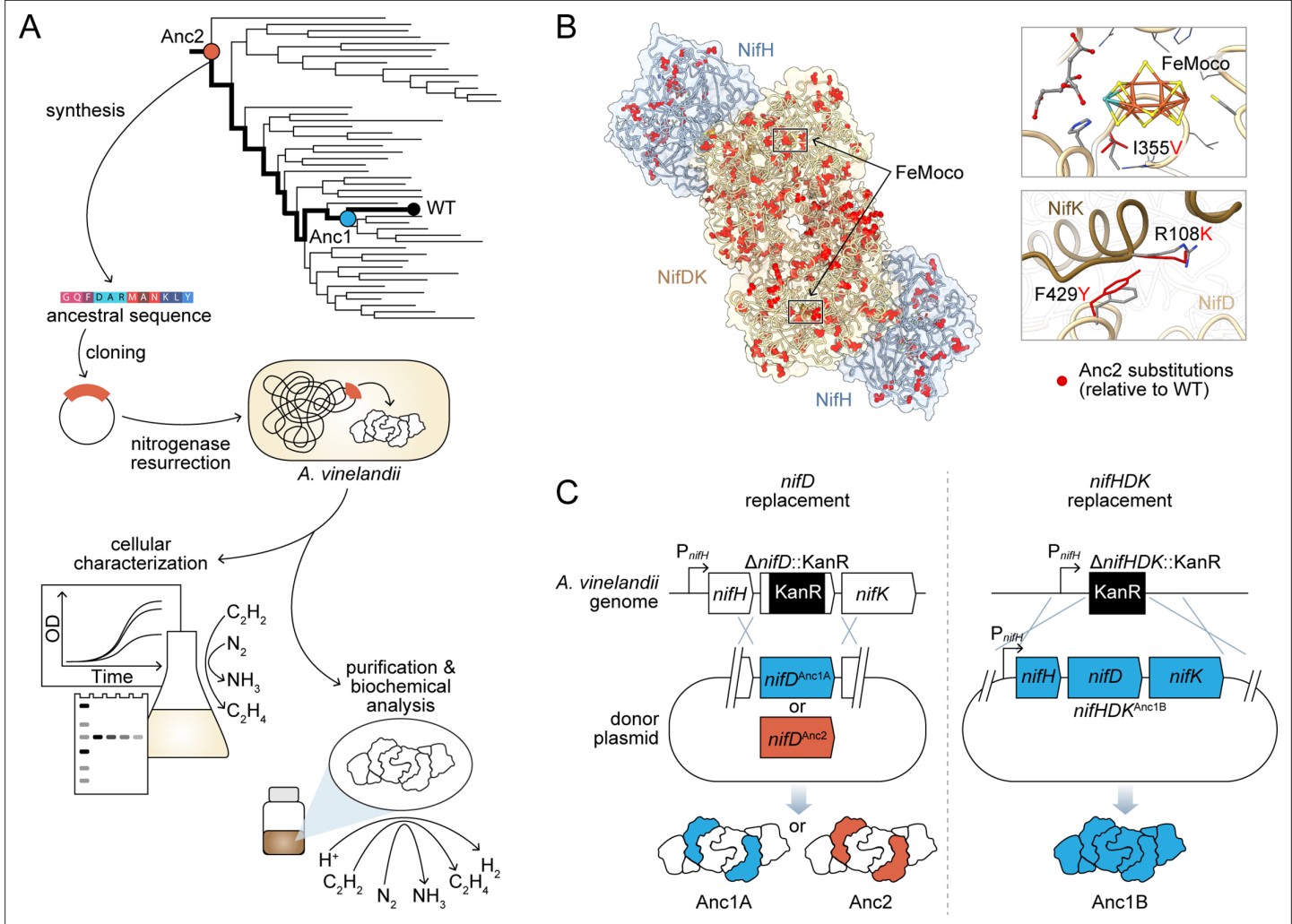

**Figure 1.** Engineering strategy for ancestral nitrogenase resurrection. (**A**) Experimental pipeline for nitrogenase resurrection in *A. vinelandii* and subsequent characterization, as described in the main text. (**B**) Structural overview of ancestral nitrogenases reconstructed in this study. Homology models (template PDB 1M34) of Anc1B NifH and NifDK proteins are shown with ancestral substitutions (relative to WT) highlighted in red. Select substitutions at relatively conserved sites in proximity to FeMoco (NifD, I355V) and the NifD:NifK interface (NifD, F429Y; NifK, R108K) are displayed in the insets. (**C**) Parallel genome engineering strategies were executed in this study, involving both ancestral replacement of only *nifD* (Anc1A and Anc2) and replacement of *nifHDK* (Anc1B). 'P$_{nifH}$': *nifH* promoter, 'KanR': kanamycin resistance cassette. *Anc1A and Anc1B were each reconstructed from equivalent nodes of alternate phylogenies (see Materials and methods).

Purified ancestral enzymes exhibit the specific N$_2$ reduction mechanism retained by their studied, extant counterparts, and maintain the same catalytic selectivity between N$_2$ and protons. Thus, the core strategy for biological nitrogen fixation is conserved across the investigated timeline. Our paleo-molecular approach opens a new route to study the ancient functionality and evolution of nitrogenases both deeper in its ancestry and within the broader context of its supporting, cellular machinery.

## Results

### A resurrection strategy for ancestral nitrogenases

We designed an engineering pipeline for the resurrection and experimental characterization of ancestral nitrogenases (*Figure 1A*). In this scheme, phylogenetically inferred, ancestral nitrogenase genes are synthesized and engineered into the genome of a modern diazotrophic bacterium, enabling the assessment of in vivo nitrogenase activity and expression in parallel with biochemical analysis of the purified enzyme. The engineering and functional assessment of ancestral nitrogenases required a suitable

diazotrophic microbial host, owing to challenges associated with nitrogenase heterologous expression (*Vicente and Dean, 2017*). We selected the obligately aerobic gammaproteobacterium, *A. vinelandii* (strain 'DJ'), an ideal experimental model due to its genetic tractability and the availability of detailed studies on the genetics and biochemistry of its nitrogen fixation machinery (*Noar and Bruno-Bárcena, 2018*). We specifically targeted the extant *A. vinelandii* molybdenum-dependent ('Mo-;) nitrogenase (hereafter referred to as wild-type, 'WT'), which is the best-studied isozyme (*Seefeldt et al., 2020*) relative to the *A. vinelandii* vanadium ('V-') and iron ('Fe-') dependent nitrogenases. The WT Mo-nitrogenase complex comprises multiple subunits, NifH, NifD, and NifK (encoded by *nifHDK* genes), which are arranged into two catalytic components: a NifH homodimer and a NifDK heterotetramer (*Figure 1B*). During catalysis, both components transiently associate to transfer one electron from NifH to NifDK and subsequently dissociate. Transferred electrons accumulate at the active-site Mo-containing metallocluster ('FeMoco') housed within the NifD subunits for reduction of the $N_2$ substrate to $NH_3$.

To infer Mo-nitrogenase ancestors, we built a maximum-likelihood nitrogenase phylogeny from a concatenated alignment of NifHDK amino acid sequences (*Figure 2A*; *Figure 2—figure supplement 1*). The phylogeny contains 385 sets of homologs representative of known nitrogenase molecular sequence diversity (including Mo-, V-, and Fe-nitrogenases), and is rooted by dark-operative protochlorophyllide oxidoreductase proteins classified within the nitrogenase superfamily (*Ghebreamlak and Mansoorabadi, 2020*). For this study, we selected ancestors that fall within the direct evolutionary lineage of *A. vinelandii* WT (*Figure 2B*), 'Anc1' and 'Anc2' (listed in order of increasing age), having ~90% and ~85% amino acid sequence identity to WT across the full length of their concatenated NifHDK proteins, respectively (*Figure 2C*; *Supplementary file 1a*). A relatively conservative percentage identity threshold was chosen based on prior studies benchmarking the functional expression of ancestral elongation factor proteins in *Escherichia coli* (*Kacar et al., 2017a*) and *Synechococcus elongatus* (*Kędzior et al., 2022*). The high-dimensional, nitrogenase protein sequence space occupied by both extant and ancestral homologs is visualized in two dimensions in *Figure 2D* by machine-learning embeddings (see Materials and methods). This analysis highlights the swath of sequence space targeted here, as well as that made accessible by the resurrection of nitrogenase ancestors more broadly. WT, Anc1, and Anc2 lie within a Mo-nitrogenase clade (previously termed 'Group I' [*Raymond et al., 2004*]) that contains homologs from diverse aerobic and facultatively anaerobic taxa, including proteobacteria and cyanobacteria (*Figure 2A*). A maximum age constraint of ~2.5 Ga for Group I nitrogenases (and thus for both Anc1 and Anc2) can be reasoned based on the timing of the Great Oxidation Event (*Lyons et al., 2014*) and downstream emergence of aerobic taxa represented nearly exclusively within this clade.

Residue-level differences between ancestral and WT nitrogenases ('ancestral substitutions') are broadly distributed along the length of each ancestral sequence (*Figure 1B*; *Figure 2—figure supplement 2*). An ancestral substitution proximal to the active-site FeMoco metallocluster lies within a loop considered important for FeMoco insertion (*Dos Santos et al., 2004*) (NifD I355V; residue numbering from WT) and is observed across all targeted NifD ancestors (*Figure 1B*). Other ancestral substitutions are notable for their location at relatively conserved residue sites (assessed by ConSurf *Ashkenazy et al., 2016*; *Figure 2—figure supplement 3*; see Materials and methods) and/or within subunit interfaces, including two at the NifD:NifK interface that are proximal to one another, F429Y (NifD) and R108K (NifK). The Anc2 NifD protein contains five more ancestral substitutions at conserved sites than the younger Anc1 NifD protein. In all studied ancestors, the C275 and H442 FeMoco ligands, as well as other strictly conserved nitrogenase residues, are retained.

Phylogenetic analysis informs the compatibility of selected ancestors in extant microbial hosts. Extant nitrogenases within the Group I nitrogenase clade (which include Anc1 and Anc2 descendants) are associated with numerous accessory genes likely recruited to optimize the synthesis and regulation of the oxygen-sensitive nitrogenase for aerobic or facultative metabolisms (*Boyd et al., 2015*). For example, in addition to the structural *nifHDK* genes, *A. vinelandii* WT is assembled and regulated with the help of >15 additional *nif* genes. Likewise, the extant descendants of Anc1 and Anc2 are primarily aerobic or facultative proteobacteria and are thus associated with higher complexity *nif* gene clusters (*Figure 2B*). We hypothesized that the likely oxygen-tolerant, ancient proteobacterium harboring these ancestral nitrogenases were similar in *nif* cluster complexity to extant *A. vinelandii*. Thus, we predicted that the *nif* accessory genes present in *A. vinelandii* would support the functional expression of resurrected nitrogenase ancestors.

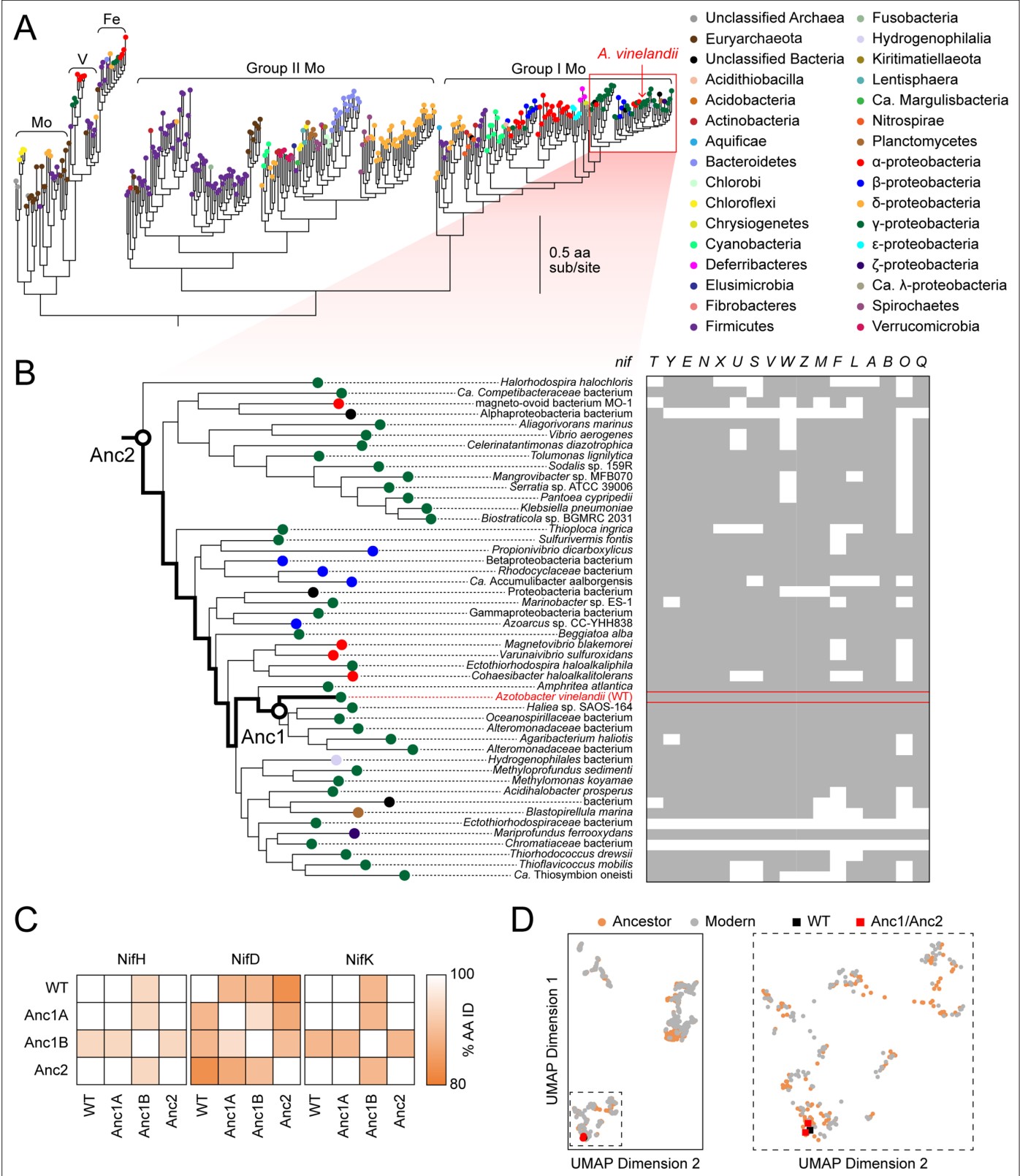

**Figure 2.** Phylogenetic and genomic context of resurrected ancestral nitrogenases. (**A**) Maximum-likelihood phylogenetic tree from which ancestral nitrogenases were inferred. Extant nodes are colored by microbial host taxonomic diversity. Red box highlights the clade targeted in this study and depicted in (**B**). Tree shown was used to infer Anc1A and Anc2 sequences (an alternate tree was used for Anc1B inference; see **Materials and methods**). (**B**) *nif* gene cluster complexity within the targeted nitrogenase clade. Presence and absence of each *nif* gene are indicated by gray and white colors,

*Figure 2 continued on next page*

Figure 2 continued

respectively. Because some homologs for phylogenetic analysis were obtained from organisms lacking fully assembled genomes, the absence of accessory *nif* genes may result from missing genomic information. (**C**) Amino acid sequence identity matrix of nitrogenase protein subunits harbored by WT and engineered *A. vinelandii* strains. (**D**) Extant and ancestral nitrogenase protein sequence space visualized by machine-learning embeddings, with the resulting dimensionality reduced to two-dimensional space. UMAP dimension axes are in arbitrary units. The field demarcated by dashed lines in the left plot is expanded on the right plot.

The online version of this article includes the following figure supplement(s) for figure 2:

**Figure supplement 1.** Maximum-likelihood phylogenies built from nitrogenase NifHDK homologs.

**Figure supplement 2.** Alignment of WT and ancestral NifHDK nitrogenase proteins.

**Figure supplement 3.** WT and ancestral NifD alignment with sitewise ConSurf conservation scores (see Materials and Methods).

To gauge the degree of compatibility between ancestral and extant nitrogenase proteins and maximize the chance of recovering functional nitrogenase variants, we executed two parallel genome engineering strategies. First, we constructed *A. vinelandii* strains harboring only the ancestral *nifD* gene from both targeted ancestral nodes ('Anc1A' and 'Anc2'), thereby expressing 'hybrid ancestral-WT' nitrogenase complexes (*Figure 1C*). This strategy is similar to in vitro 'cross-reaction' studies that have evaluated the compatibility of nitrogenase protein components from differing host taxa (*Smith et al., 1976*). Second, we constructed a strain harboring all Anc1 *nifHDK* genes, expressing a fully ancestral nitrogenase complex ('Anc1B'; sequence reconstructed from a node equivalent to Anc1A from an alternate phylogeny, see **Materials and methods**). *A. vinelandii* strains were constructed by markerless genomic integration of ancestral nitrogenase genes, as described in **Materials and methods**.

## Ancestral nitrogenases enable diazotrophic microbial growth

All *A. vinelandii* constructs harboring ancestral genes enabled diazotrophic growth in molybdenum-containing, nitrogen-free media. All strains had comparable doubling times to WT during the exponential phase ($p > 0.05$; *Figure 3A and B*). The only significant difference among strains was a ~14 hr increase in the lag phase of strain Anc2 relative to WT, harboring the oldest nitrogenase ancestor ($p \approx$ 2e-7). We did not detect growth under the same conditions for a control Δ*nifD* strain (DJ2278, see *Supplementary file 1c*). This result confirmed that the growth observed for ancestral strains did not stem from leaky expression of the alternative, V- or Fe-dependent nitrogen fixation genes in *A. vinelandii,* which were left intact.

An acetylene reduction assay was performed to measure cellular nitrogenase activity in engineered strains. This assay quantifies the reduction rate of the non-physiological substrate acetylene ($C_2H_2$) to ethylene ($C_2H_4$) (*Hardy et al., 1968*), here normalized to total protein content. *A. vinelandii* strains harboring only ancestral *nifD* (Anc1A, Anc2) exhibited mean $C_2H_2$ reduction rates of ~5–6 μmol $C_2H_4$/mg total protein/hr, ~40–45% that of WT ($p \approx$ 6e-4 and $p \approx$ 3e-4, respectively) (*Figure 3C*). Strain Anc1B, harboring ancestral *nifHDK,* exhibited a mean acetylene reduction rate of ~9 μmol $C_2H_4$/mg total protein/hr, ~70% that of WT ($p \approx$ 3e-2).

The phenotypic variability we observed among engineered and WT *A. vinelandii* strains might result both from differences in nitrogenase expression and nitrogenase activity. To provide insights into these disparate effects, we quantified nitrogenase protein expression in engineered strains by immunodetection of ancestral and WT Strep-tagged NifD proteins (the latter from strain DJ2102, see *Supplementary file 1c*) and did not conclusively detect significant differences in protein quantity relative to WT ($p > 0.05$; *Figure 3D*).

## Purified ancestral nitrogenases conserve extant N$_2$ reduction mechanisms and efficiency

Ancestral nitrogenase NifDK protein components were expressed and purified for biochemical characterization. All ancestral NifDK proteins were assayed with WT NifH protein (ancestral NifH proteins were not purified) for reduction of $H^+$, $N_2$, and $C_2H_2$. Ancestors were found to reduce all three substrates in vitro, supporting the cellular-level evidence of ancestral nitrogenase activity (*Figure 4A*).

We investigated whether the ancestral nitrogenases studied here would exhibit the general mechanism for $N_2$ binding and reduction that has been observed for the studied, extant nitrogenase isozymes of *A. vinelandii* (Mo, V, and Fe) (*Harris et al., 2019*; *Harris et al., 2022*). This mechanism involves the

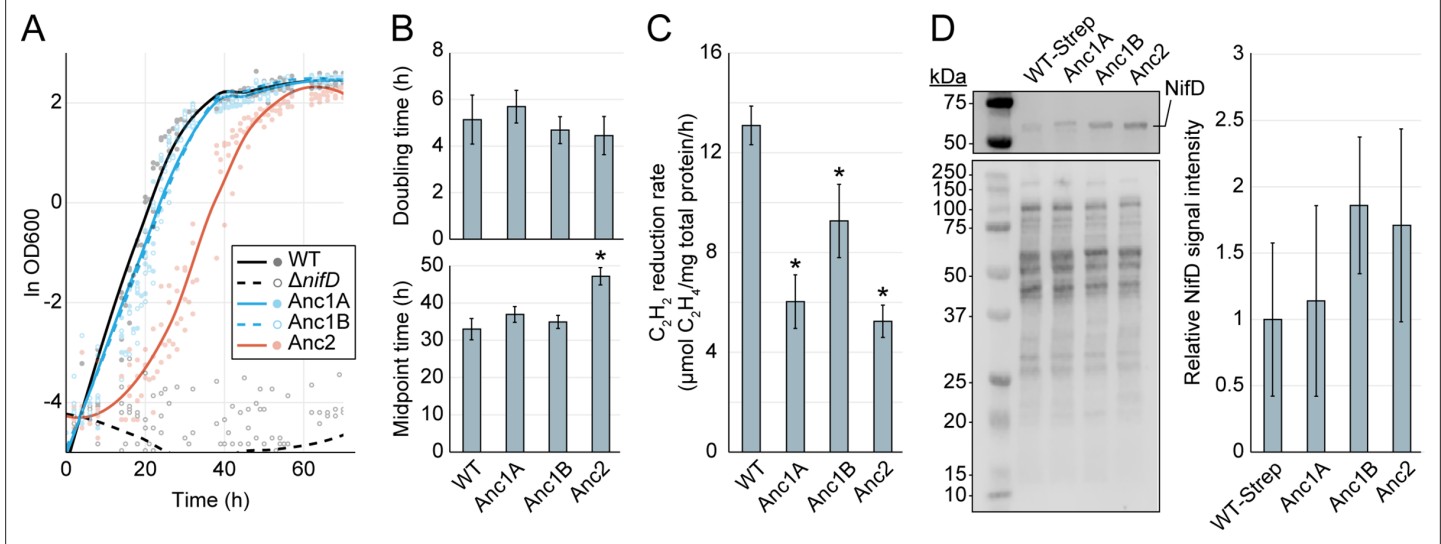

**Figure 3.** Cellular-level characterization of ancestral nitrogenase activity and expression. (**A**) Diazotrophic growth curves of *A. vinelandii* strains measured by the optical density at 600 nm ('OD600'). A smoothed curve is shown alongside individual data points obtained from five biological replicates per strain. The non-diazotrophic DJ2278 (Δ*nifD*) strain was used as a negative control. (**B**) Mean doubling and midpoint times of *A. vinelandii* strains, calculated from data in (**A**). (**C**) In vivo acetylene ($C_2H_2$) reduction rates quantified by the production of ethylene ($C_2H_4$). Bars represent the mean of biological replicates (n=3) per strain. (**D**) Immunodetection and protein quantification of Strep-II-tagged WT ('WT-Strep,' strain DJ2102) and ancestral NifD. Top gel image shows Strep-II-tagged NifD proteins detected by anti-Strep antibody and bottom gel image shows total protein stain. Plot displays relative immunodetected NifD signal intensity normalized to total protein intensity and expressed relative to WT. Bars in the plot represent the mean of biological replicates (n=3) per strain. (**B–D**) Error bars indicate ±1 SD and asterisks indicate p<.01 (one-way ANOVA, post-hoc Tukey HSD) compared to WT or WT-Strep.

The online version of this article includes the following source data for figure 3:

**Source data 1.** Source Excel file for diazotrophic growth curve data and statistical analyses.

**Source data 2.** Source Excel file for in vivo acetylene reduction assay data and statistical analyses.

**Source data 3.** Source Excel file for NifD protein densitometry data and statistical analyses.

**Source data 4.** Zip archive of Western blot image data (total protein stain, all strains, replicate 1), containing labeled and unlabeled image files.

**Source data 5.** Zip archive of Western blot image data (all strains, replicate 1), containing labeled and unlabeled image files.

**Source data 6.** Zip archive of Western blot image data (total protein stain, all strains, replicate 2), containing labeled and unlabeled image files.

**Source data 7.** Zip archive of Western blot image data (all strains, replicate 2), containing labeled and unlabeled image files.

**Source data 8.** Zip archive of Western blot image data (total protein stain, all strains, replicate 3), containing labeled and unlabeled image files.

**Source data 9.** Zip archive of Western blot image data (all strains, replicate 3), containing labeled and unlabeled image files.

accumulation of four electrons/protons on the active-site cofactor as metal-bound hydrides, generating the $E_4(4 H)$ state (*Figure 4B*). Once generated, $N_2$ can bind to the $E_4(4 H)$ state through a reversible reductive elimination/oxidative addition (*re/oa*) mechanism, which results in the release (*re*) of a single molecule of hydrogen gas ($H_2$). $N_2$ binding is reversible in the presence of sufficient $H_2$, which displaces bound $N_2$ and results in the reformation of $E_4(4 H)$ with two hydrides (*oa*). Thus, a classic test of the (*re/oa*) mechanism is the ability of $H_2$ to inhibit $N_2$ reduction. We observed that the reduction of $N_2$ to $NH_3$ for all nitrogenase ancestors was inhibited in the presence of $H_2$, indicating that the ancestors follow the same mechanism of $N_2$ binding determined for extant enzymes (*Figure 4C*).

In the event the $E_4(4 H)$ state fails to capture $N_2$, nitrogenases will simply produce $H_2$ from the $E_4(4 H)$ state to generate the $E_2(2 H)$ state. The ratio of $H_2$ formed to $N_2$ reduced ($H_2/N_2$) can be used as a measure of the efficiency of nitrogenases in using ATP and reducing equivalents for $N_2$ reduction. The stoichiometric minimum of the mechanism is $H_2/N_2=1$. Experimentally (under 1 atm $N_2$), a ratio of ~2 is seen for Mo-nitrogenase and ~5 and~7 for V- and Fe-nitrogenase, respectively (*Harris et al., 2019*). $H_2/N_2$ values for all ancestors under 1 atm $N_2$ was ~2, similar to extant Mo-nitrogenase (*Figure 4D*).

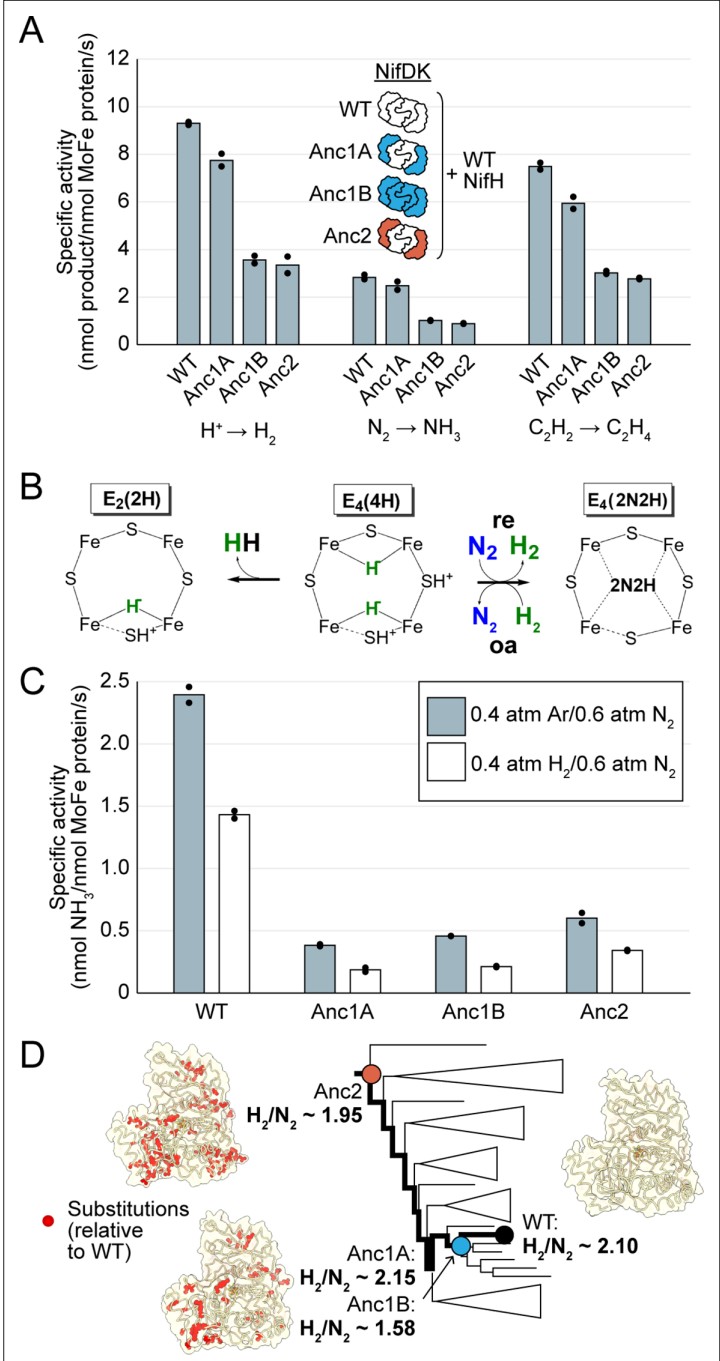

**Figure 4.** In vitro analyses of ancestral nitrogenase activity profiles and mechanism. All measurements were obtained from assays using purified NifDK assayed with WT NifH. (**A**) Specific activities were measured for $H^+$, $N_2$, and $C_2H_2$ substrates. (**B**) Partial schematic of the reductive-elimination $N_2$-reduction mechanism of nitrogenase is shown above, centering on the $N_2$-binding $E_4(4 H)$ state of FeMoco (see main text for discussion) (***Harris et al., 2019***). (**C**) Inhibition of $N_2$ reduction by $H_2$, evidencing the mechanism illustrated in (**B**). (**D**) Catalytic efficiencies of ancestral nitrogenases, described by the ratio of formed $H_2$ to reduced $N_2$ ($H_2/N_2$), mapped across the targeted phylogenetic clade. NifD homology models (PDB 1M34 template) are displayed with ancestral substitutions highlighted in red. (**A,C**) Bars represent the mean of independent experiments (n=2) with individual data points shown as black circles.

The online version of this article includes the following source data and figure supplement(s) for figure 4:

**Source data 1.** Source Excel file for nitrogenase in vitro activity data.

*Figure 4 continued on next page*

*Figure 4 continued*

**Source data 2.** Source Excel file for nitrogenase in vitro $H_2$ inhibition data.

**Figure supplement 1.** SDS-PAGE of purified WT and ancestral NifDK proteins.

**Figure supplement 1—source data 1.** Zip archive of SDS-PAGE image data, containing labeled and unlabeled image files.

## Discussion

In this study, we leverage a new approach to investigate ancient nitrogen fixation by the resurrection and functional assessment of ancestral nitrogenase enzymes. We demonstrate that engineered *A. vinelandii* cells can reduce $N_2$ and $C_2H_2$ and exhibit diazotrophic growth rates comparable to WT, though we observe that the oldest ancestor, Anc2, has a significantly longer lag phase. Purified nitrogenase ancestors are active for the reduction of $H^+$, $N_2$, and $C_2H_2$, while maintaining the catalytic efficiency (described by the $H_2/N_2$ ratio) of WT enzymes. Our results also show that ancestral $N_2$ reduction is inhibited by $H_2$, indicating an early emergence of the reductive-elimination $N_2$-reduction mechanism preserved by characterized, extant nitrogenases (*Harris et al., 2019*; *Harris et al., 2022*). These properties are maintained despite substantial residue-level changes to the peripheral nitrogenase structure (including relatively conserved sites), as well as a handful within the active-site or protein-interface regions within the enzyme complex.

It is important to consider that the nitrogenase ancestors resurrected here represent hypotheses regarding the true ancestral state. Uncertainty underlying ancestral reconstructions might derive from incomplete extant molecular sequence data, as well as incorrect assumptions associated with the implemented evolutionary models (*Garcia and Kaçar, 2019*). For example, a complicating feature of nitrogenase evolution is that it has been shaped significantly by horizontal gene transfer (*Raymond et al., 2004*; *Parsons et al., 2021*), which in certain cases has led to different evolutionary trajectories of individual nitrogenase structural genes. Specifically, certain H-subunit genes of the V-nitrogenase appear to have different evolutionary histories relative to their DK components (*Raymond et al., 2004*). However, since our study targets a Mo-nitrogenase lineage, we do not expect horizontal transfer to be a significant source of uncertainty in our reconstructions.

The $N_2$-reduction activity of nitrogenase ancestors suggests that the required protein-protein interactions—both between subunits that comprise the nitrogenase complex as well as those required for nitrogenase assembly in *A. vinelandii*—and metallocluster interactions are sufficiently maintained for primary function. Still, our results reveal the degree to which the organism-level phenotype of host strains can be perturbed by varying both the number and age of ancestral subunits. Importantly, these changes appear to impact phenotypic properties in complex ways, representative of the type of cellular constraints on nitrogenase evolution that would be unobservable through an in vitro study alone. For example, we observed comparable growth characteristics of strains harboring single (Anc1A, ancestral NifD) versus multiple (Anc1B, ancestral NifHDK) ancestral subunits of equivalent age, whereas the lag phase of Anc2 hosting a single, older subunit (ancestral NifD) was increased. Here, growth is sensitive to older, ancestral substitutions in a single subunit while permissive of more recent ancestral substitutions across one or more subunits within the nitrogenase complex. However, a different pattern is observed across in vivo acetylene reduction rates. These are most negatively impacted relative to WT in strains with a single NifD ancestor (Anc1A, Anc2), whereas rates are more modestly decreased in a strain with a complete, ancestral NifHDK complex (Anc1A). These results suggest that ancestral subunits of equivalent age have greater compatibility and yield greater in vivo activity compared to subunits of disparate ages, perhaps owing to modified protein interactions within the nitrogenase complexes The discrepancy between in vivo activity and growth characteristics may also be attributable to impacted cellular processes external to the biochemical properties of the nitrogenase complex itself, and yet nevertheless vital in determining the overall fitness of the host organism. Finally, though we do not detect significant differences in ancestral protein expression here, it is possible that phenotypic outcomes of future reconstructions might be impacted by perturbed expression levels (e.g. *Kędzior et al., 2022*; *Garcia et al., 2021b*). To what degree these expression levels are representative of the ancestral state and impact the phenotypic property of interest should be considered in future work.

That nitrogenase ancestors perform the reductive-elimination $N_2$-reduction mechanism—as the distantly related (*Garcia et al., 2020*), extant Mo-, V-, and Fe-nitrogenases of *A. vinelandii* do today (*Harris et al., 2019*)—likely indicates that this enzymatic characteristic was set early in nitrogenase evolutionary history and sustained through significant past environmental change (*Lyons et al., 2014*; *Som et al., 2016*; *Catling and Zahnle, 2020*) and ecological diversification (*Boyd et al., 2015*; *Zehr et al., 2003*). It is possible that life's available strategies for achieving $N_2$ reduction may be fundamentally limited, and that a defining constraint of nitrogenase evolution has been the preservation of the same $N_2$ reduction mechanism across shifting selective pressures. For example, in the acquisition of V- and Fe-dependence from Mo-dependent ancestors (*Garcia et al., 2020*), nitrogenases may have required substantial sequence and structural changes (*Sippel and Einsle, 2017*; *Eady, 1996*) in order to facilitate reductive elimination given a different active-site metallocluster. It is also possible that alternate strategies for biological nitrogen fixation evolved early in the history of life and were subsequently outcompeted, leaving no trace of their existence in extant microbial genomes. Why these alternate possibilities were evidently not explored by nature to the same degree remains an open question, particularly given the several abiotic mechanisms for nitrogen fixation (*Cherkasov et al., 2015*; *Dörr et al., 2003*; *Yung and McElroy, 1979*) and the multiple biological pathways for another, globally significant metabolism, carbon fixation (*Garcia et al., 2021a*). Because our paleomolecular approach is ultimately informed by extant sequence data, it cannot directly evaluate extinct sequences that, for instance, due to contingency or entrenchment, did not persist and become preserved in extant microbial genomes. Nevertheless, evolutionarily informed studies of nitrogenase functionality that define the sequence-function space of this enzyme family will provide a foundation for laboratory efforts aimed toward exploring alternate scenarios. Future work that explores deeper into nitrogenase evolutionary history (and across extant and ancestral nitrogenase sequence space, as charted here (*Figure 2D*)) will clarify the degree of functional constraint exhibited by the nitrogenase family, both past and present.

## Conclusion

Broadening the historical level of analysis beyond a single enzyme to the organism level is necessary to generate comprehensive insights into the evolutionary history and engineering potential of nitrogen fixation. Paleomolecular work that has expanded toward the systems-level investigation of early-evolved, crucial metabolic pathways remains in its infancy, despite the potential for provocative connections between molecular-scale innovations and planetary history (*Garcia and Kaçar, 2019*; *Kędzior et al., 2022*). Our results highlight the evolutionary conservation of a critical metabolic pathway that has shaped the biosphere over billions of years, as well as establish the tractability of leveraging phylogenetic models to carry out extensive, empirical manipulations of challenging enzymatic systems and their microbial hosts. Building on the empirical framework presented here will illuminate the evolutionary design principles behind ancient metabolic systems more broadly as well as leverage these histories to understand how key enzymes that allowed organisms to access nitrogen from the atmosphere evolved.

# Materials and methods

## Key resources table

| Reagent type (species) or resource | Designation | Source or reference | Identifiers | Additional information |
|---|---|---|---|---|
| strain, strain background (*A. vinelandii*) | DJ | DOI:10.1128/JB.00504–09 | n/a | Dennis Dean, Virginia Tech; Wild-type (WT); Nif+ |
| genetic reagent (*A. vinelandii*) | DJ2102 | DOI:10.1016/bs.mie.2018.10.007 | n/a | Dennis Dean, Virginia Tech; Strep-tagged WT NifD; Nif+ |
| genetic reagent (*A. vinelandii*) | DJ2278 | Other | n/a | Dennis Dean, Virginia Tech; Δ*nifD*::KanR; Nif- |
| genetic reagent (*A. vinelandii*) | DJ884 | Other | n/a | Dennis Dean, Virginia Tech; *nifD*R187I mutant; Nif+(slow); overexpresses NifH |
| genetic reagent (*A. vinelandii*) | AK022 | This paper | n/a | Δ*nifHDK*::KanR; Nif- |

*Continued on next page*

*Continued*

| Reagent type (species) or resource | Designation | Source or reference | Identifiers | Additional information |
|---|---|---|---|---|
| genetic reagent (*A. vinelandii*) | AK013 | This paper | n/a | 'Anc1A'; Δ*nifD::nifD*^Anc1A^; Nif+ |
| genetic reagent (*A. vinelandii*) | AK023 | This paper | n/a | 'Anc1B'; Δ*nifHDK::nifHDK*^Anc1B^; Nif+ |
| genetic reagent (*A. vinelandii*) | AK014 | This paper | n/a | 'Anc2'; Δ*nifD::nifD*^Anc2^; Nif+ |
| antibody | StrepMAB-Classic (Mouse monoclonal) | IBA Lifesciences | Cat# 2-1507-001, RRID: AB_513133 | WB (1:5000) |
| recombinant DNA reagent | pAG25 | This paper | n/a | KanR cassette (APH(3')-I gene)+400 bp *nifHDK* flanking homology regions, synthesized into XbaI/KpnI sites in pUC19; used to construct strain AK022 from DJ |
| recombinant DNA reagent | pAG13 | This paper | n/a | *nifD*^Anc1A^ + 400-bp *nifD* flanking homology regions, synthesized into XbaI/KpnI sites in pUC19; used to construct strain Anc1A from AK022 |
| recombinant DNA reagent | pAG19 | This paper | n/a | *nifHDK*^Anc1B^ + 400-bp *nifHDK* flanking homology regions, synthesized into XbaI/KpnI sites in pUC19; used to construct strain Anc1B from AK022 |
| recombinant DNA reagent | pAG14 | This paper | n/a | *nifD*^Anc2^ +400 bp *nifD* flanking homology regions, synthesized into XbaI/KpnI sites in pUC19; used to construct strain Anc2 from AK022 |
| sequence-based reagent | 306_nifH_F | This paper | PCR primers | GCCGAACGTTCAAGTGGAAA |
| sequence-based reagent | 307_nifH_R | This paper | PCR primers | AGAGCCAATCTGCCCTGTC |
| sequence-based reagent | 308_nifD_F | This paper | PCR primers | CACCCGTTACCCGCATATGA |
| sequence-based reagent | 309_nifD_R | This paper | PCR primers | ACTCATCTGTGAACGGCGTT |
| sequence-based reagent | 310_nifK_F | This paper | PCR primers | GCTAACGCCGTTCACAGATG |
| sequence-based reagent | 311_nifK_R | This paper | PCR primers | TCAGTTGGCCTTCGTCGTTG |
| software, algorithm | MAFFT | MAFFT | RRID:SCR_011811 | |
| software, algorithm | trimAl | trimAl | RRID:SCR_017334 | |
| software, algorithm | IQ-TREE | IQ-TREE | RRID:SCR_017254 | |
| software, algorithm | RAxML | RAxML | RRID:SCR_006086 | |
| software, algorithm | PAML | PAML | RRID:SCR_014932 | |
| software, algorithm | MODELLER | MODELLER | RRID:SCR_008395 | |
| software, algorithm | ChimeraX | ChimeraX | RRID:SCR_015872 | |
| software, algorithm | Growthcurver | Growthcurver | n/a | R package |

## Nitrogenase ancestral sequence reconstruction and selection

The nitrogenase protein sequence dataset was assembled by BLASTp (*Camacho et al., 2009*) search of the NCBI non-redundant protein database (accessed August 2020) with *A. vinelandii* NifH (WP_012698831.1), NifD (WP_012698832.1), and NifK (WP_012698833.1) queries and a 1e-5 Expect value threshold (*Supplementary file 1b*). BLASTp hits were manually curated to remove partially sequenced, misannotated, and taxonomically overrepresented homologs. BLASTp hits included protein sequences from homologous Mo-, V-, and Fe-nitrogenase isozymes (*Garcia et al., 2020*). H-, D, and K-subunit sequences from these isozymes were individually aligned by MAFFT v7.450 (*Katoh and Standley, 2013*) and concatenated along with outgroup dark-operative protochlorophyllide oxidoreductase sequences (Bch/ChlLNB). The final dataset included 385 nitrogenase sequences and 385 outgroup sequences. For sequences used to construct Anc1A and Anc2 (internal nodes #960 and #929, respectively), tree reconstruction (using a trimmed alignment generated by trimAl v1.2 [*Capella-Gutiérrez et al., 2009*]), and ancestral sequence inference (using the initial untrimmed alignment) were both performed by RAxML v8.2.10 (*Stamatakis, 2014*) with the LG +G + F evolutionary

model (model testing performed by the ModelFinder *Kalyaanamoorthy et al., 2017* in the IQ-TREE v.1.6.12 package [*Nguyen et al., 2015*]).

Due to concerns that RAxML v.8.2 does not implement full, marginal ancestral sequence reconstruction as described by *Yang et al., 1995*, we performed a second phylogenetic analysis as follows. The extant sequence dataset described above was realigned by MAFFT (untrimmed) and tree reconstruction was again performed by RAxML. Ancestral sequence reconstruction was instead performed by PAML v4.9j (*Yang, 2007*) using the LG +G + F model. From this second reconstruction, Anc1B (internal node #1312), equivalent to Anc1A, was selected for experimental analysis. Anc1B and Anc1A have identical sets of descendent homologs, and their NifD proteins are 95% identical (*Figure 2B*).

Only the ancestral sequences inferred with the most probable residue at each protein site were considered for this study (mean posterior probabilities of targeted nitrogenase subunits range from 0.95 to 0.99; see *Supplementary file 1a*). All ancestral sequences were reconstructed from well-supported clades (SH-like aLRT = 99–100 [*Anisimova and Gascuel, 2006*]).

## Ancestral nitrogenase structural modeling and sequence analysis

Structural homology models of ancestral sequences were generated by MODELLER v10.2 (*Webb and Sali, 2016*) using PDB 1M34 as a template for all nitrogenase protein subunits and visualized by ChimeraX v1.3 (*Pettersen et al., 2021*).

Extant and ancestral protein sequence space was visualized by machine-learning embeddings, where each protein embedding represents protein features in a fixed-size, multidimensional vector space. The analysis was conducted on concatenated (HDK) nitrogenase protein sequences in our phylogenetic dataset. The embeddings were obtained using the pre-trained language model ESM2 (*Lin et al., 2022*; *Rives et al., 2021*), a transformer architecture trained to reproduce correlations at the sequence level in a dataset containing hundreds of millions of protein sequences. Layer 33 of this transformer was used, as recommended by the authors. The resulting 1024 dimensions were reduced by UMAP (*McInnes et al., 2020*) for visualization in a two-dimensional space.

Protein site-wise conservation analysis was performed using the Consurf server (*Ashkenazy et al., 2016*). An input alignment containing only extant, Group I Mo-nitrogenases was submitted for analysis under default parameters. Conserved sites were defined by a Consurf conservation score >7.

## *A. vinelandii* strain engineering

Nucleotide sequences of targeted ancestral nitrogenase proteins were codon-optimized for *A. vinelandii* by a semi-randomized strategy that maximized ancestral nucleotide sequence identity to WT genes. Ancestral and WT protein sequences were compared using the alignment output of ancestral sequence reconstruction (RAxML or PAML). For sites where the ancestral and WT residues were identical, the WT codon was assigned. At sites where the residues were different, the codon was assigned randomly, weighted by *A. vinelandii* codon frequencies (Codon Usage Database, https://www.kazusa.or.jp/codon/). Nucleotide sequences were synthesized into XbaI/KpnI sites of pUC19 vectors (unable to replicate in *A. vinelandii*) (Twist Bioscience; GenScript). Inserts were designed with 400-base-pair flanking regions for homology-directed recombination at the relevant *A. vinelandii* nif locus. An 'ASWSHPQFEK' Strep-II-tag was included at the N-terminus of each synthetic *nifD* gene for downstream NifD immunodetection and NifDK affinity purification. See *Supplementary file 1c* for a list of strains and plasmids used in this study.

Engineering of *A. vinelandii* strains used established methods, following *Dos Santos, 2019*. *A. vinelandii* WT ('DJ'), DJ2278 (Δ*nifD*::KanR), DJ2102 (Strep-II-tagged WT NifD), and DJ884 (NifH-overexpression mutant) strains were generously provided by Dennis Dean (Virginia Tech) (*Supplementary file 1c*). Strains Anc1A and Anc2 were constructed from the DJ2278 parent strain via transformation with plasmids pAG13 and pAG19, respectively (*Supplementary file 1c*). For the construction of strain Anc2, we first generated a Δ*nifHDK* strain, AK022, by transforming the DJ strain with pAG25. Genetic competency was induced by subculturing relevant parent strains in Mo- and Fe-free Burk's medium (see below). Competent cells were transformed with at least 1 µg of donor plasmid. Transformants were screened on solid Burk's medium for the rescue of the diazotrophic phenotype ('Nif+') and loss of kanamycin resistance, followed by Sanger sequencing of the PCR-amplified *nifHDK* cluster (see *Supplementary file 1d* for a list of primers). Transformants were passaged at least three times to ensure phenotypic stability prior to storage at –80 °C in phosphate buffer containing 7% DMSO.

### A. vinelandii culturing and growth analysis

*A. vinelandii* strains were grown diazotrophically in nitrogen-free Burk's medium (containing 1 µM $Na_2MoO_4$) at 30 °C and agitated at 300 rpm. To induce genetic competency for transformation experiments, Mo and Fe salts were excluded. For transformant screening, kanamycin antibiotic was added to solid Burk's medium at a final concentration of 0.6 µg/mL. 50 mL seed cultures for growth rate and acetylene reduction rate quantification were grown non-diazotrophically in flasks with Burk's medium containing 13 mM ammonium acetate.

For growth rate quantification, seed cultures were inoculated into 100 mL nitrogen-free Burk's medium to an optical density of ~0.01 at 600 nm (OD600), after *Carruthers et al., 2021*, and monitored for 72 hr. Growth parameters were modeled using the R package Growthcurver (*Sprouffske and Wagner, 2016*).

### Microbial acetylene reduction assays

*A. vinelandii* seed cultures representing independent biological replicates were prepared as described above and used to inoculate 100 mL of nitrogen-free Burk's medium to an OD600 ≈ 0.01. Cells were grown diazotrophically to an OD600 ≈ 0.5, at which point a rubber septum cap was affixed to the mouth of each flask. 25 mL of headspace was removed and replaced by injecting an equivalent volume of acetylene gas. The cultures were subsequently shaken at 30 °C and agitated at 300 rpm. Headspace samples were taken after 15, 30, 45, and 60 min of incubation for ethylene quantification by a Nexis GC-2030 gas chromatograph (Shimadzu). After the 60 min incubation period, cells were pelleted at 4700 rpm for 10 min, washed once with 4 mL of phosphate buffer, and pelleted once more under the same conditions prior to storage at –80 °C. Total protein was quantified using the Quick Start Bradford Protein Assay kit (Bio-Rad) according to manufacturer instructions and a CLARIOstar Plus plate reader (BMG Labtech). Acetylene reduction rates for each replicate were normalized to total protein.

### Nitrogenase expression analysis

Strep-II-tagged NifD protein quantification was performed on all ancestral strains (Anc1A, Anc1B, Anc2) and DJ2102 (harboring Strep-II-tagged WT NifD). Diazotrophic *A. vinelandii* cultures (100 mL) representing three independent biological replicates were prepared as described above and harvested at an OD600 ≈ 1. Cell pellets were resuspended in TE lysis buffer (10 mM Tris, 1 mM EDTA, 1 mg/mL lysozyme) and heated at 95 °C for 10 min. Cell lysates were centrifuged at 5000 rpm for 15 min. Total protein in the resulting supernatant was quantified using the Pierce BCA Protein Assay kit (ThermoFisher) following manufacturer instructions. Normalized protein samples were diluted in 2×Laemmli buffer at a 1:1 (v/v) ratio prior to SDS-PAGE analysis. Proteins were transferred to nitrocellulose membranes (ThermoFisher), stained with Revert 700 Total Protein Stain (LI-COR), and imaged on an Odyssey Fc Imager (LI-COR). Membranes were then destained with Revert Destaining Solution (LI-COR) and blocked with 5% non-fat milk in PBS solution (137 mM NaCl, 2.7 mM KCl, 10 mM $Na_2HPO_4$, 1.8 mM $KH_2PO_4$.) for 1 hr at room temperature. Membranes were rinsed once with PBS-T (PBS with 0.01% Tween-20) and incubated with primary Strep-tag II antibody (Strep-MAB-Classic, IBA Lifesciences, Cat# 2-1507-001, RRID: AB_513133; 1:5000 in 0.2% BSA) for 2 hr at room temperature. Membranes were then incubated in LI-COR blocking buffer containing 1:15,000 IRDye 680RD Goat anti-Mouse (LI-COR) for 2 hr at room temperature and subsequently imaged with an Odyssey Fc Imager (LI-COR). Densitometry analysis was performed with ImageJ (*Schneider et al., 2012*), with Strep-II-tagged NifD signal intensity normalized to that of the total protein stain.

### Nitrogenase expression, purification, and biochemical characterization

Ancestral nitrogenase NifDK proteins were expressed from relevant *A. vinelandii* strains (Anc1A, Anc1B, Anc2) and purified according to previously published methods (*Jiménez-Vicente et al., 2018*) with the following modifications: cells were grown diazotrophically in nitrogen-free Burk's medium and no derepression step to a sufficient OD600 (~1.8) before harvesting. WT NifH was expressed in *A. vinelandii* strain DJ884 and purified by previously published methods (*Christiansen et al., 1998*). Protein purity was assessed at ≥95% by SDS-PAGE gel with Coomassie blue staining (*Figure 4—figure supplement 1*).

Assays were performed in 9.4 mL vials with a MgATP regeneration buffer (6.7 mM MgCl2, 30 mM phosphocreatine, 5 mM ATP, 0.2 mg/mL creatine phosphokinase, 1.2 mg/mL BSA) and 10 mM sodium

dithionite in 100 mM MOPS buffer at pH 7.0. Reaction vials were made anaerobic and relevant gases ($N_2$, $C_2H_2$, $H_2$) were added to appropriate concentrations with the headspace balanced by argon. NifDK proteins (~240 kDa) were added to 0.42 µM, the vial vented to atmospheric pressure, and the reaction initiated by the addition of NifH (~60 kDa) protein to 8.4 µM. Reactions were run, shaking, at 30 °C for 8 min and stopped by the addition of 500 µL of 400 mM EDTA pH 8.0. $NH_3$ was quantified using a fluorescence protocol (*Corbin, 1984*) with the following modifications: an aliquot of the sample was added to a solution containing 200 mM potassium phosphate pH 7.3, 20 mM o-phthalaldehyde, and 3.5 mM 2-mercaptoethanol, and incubated for 30 min in the dark. Fluorescence was measured at $\lambda_{excitation}$ of 410 nm and $\lambda_{emission}$ of 472 nm and $NH_3$ was quantified using a standard generated with $NH_4Cl$. $H_2$ and $C_2H_4$ were quantified by gas chromatography with a thermal conductivity detector (GC-TCD) and gas chromatography with a flame ionization detector (GC-FID) respectively, according to published methods (*Khadka et al., 2016*; *Yang et al., 2011*).

## Statistical analyses

Experimental data were statistically analyzed by one-way ANOVA with the post-hoc Tukey HSD test.

## Acknowledgements

We thank Dennis Dean and Valerie Cash for providing *A. vinelandii* strains DJ, DJ2278, DJ2102, and DJ884 and for guidance in genomic manipulations; Jean-Michel Ané, April MacIntyre, and Junko Maeda for guidance and instrumentation support in performing the in vivo acetylene reduction assays; Bruno Cuevas for assistance with visualizing nitrogenase sequence space; and the members of the Metal Selection and Utilization Across Eons (MUSE) Consortium for helpful suggestions and discussions. This research was supported by the National Aeronautics and Space Administration (NASA) Interdisciplinary Consortium for Astrobiology Research: Metal Utilization and Selection Across Eons, MUSE (19- ICAR19_2–0007), the University of Wisconsin-Madison College of Agricultural and Life Sciences, the NASA Postdoctoral Program (AKG), NASA Arizona Space Grant (BMC), the John Templeton Foundation (BK; 61926), the National Science Foundation (BK; 2228495), the NASA Early Career Faculty Award (BK), and the Hypothesis Fund Award (BK).

## Additional information

### Funding

| Funder | Grant reference number | Author |
|---|---|---|
| National Aeronautics and Space Administration | 19- ICAR19_2-0007 | Amanda K Garcia<br>Derek F Harris<br>Alex J Rivier<br>Brooke M Carruthers<br>Lance Seefeldt<br>Betül Kaçar<br>Azul Pinochet-Barros |
| John Templeton Foundation | 61926 | Betül Kaçar |
| National Science Foundation | 2228495 | Betül Kaçar<br>Alex J Rivier<br>Brooke M Carruthers<br>Amanda K Garcia |
| University of Wisconsin-Madison | | Betül Kaçar |
| Arizona Space Grant Consortium | | Brooke M Carruthers |
| National Aeronautics and Space Administration | 80NSSC19K1617 | Betül Kaçar |
| Hypothesis Fund | | Betül Kaçar |

| Funder | Grant reference number | Author |
|--------|------------------------|--------|

The funders had no role in study design, data collection and interpretation, or the decision to submit the work for publication.

## Author contributions

Amanda K Garcia, Conceptualization, Data curation, Formal analysis, Validation, Investigation, Visualization, Writing – original draft, Writing – review and editing; Derek F Harris, Data curation, Formal analysis, Investigation, Visualization, Writing – review and editing; Alex J Rivier, Brooke M Carruthers, Data curation, Formal analysis, Investigation; Azul Pinochet-Barros, Data curation; Lance C Seefeldt, Resources, Supervision, Writing – review and editing; Betül Kaçar, Conceptualization, Resources, Data curation, Formal analysis, Supervision, Funding acquisition, Investigation, Methodology, Writing – original draft, Project administration, Writing – review and editing

## Author ORCIDs

Amanda K Garcia (ID) http://orcid.org/0000-0002-1936-2568
Lance C Seefeldt (ID) http://orcid.org/0000-0002-6457-9504
Betül Kaçar (ID) http://orcid.org/0000-0002-0482-2357

## Decision letter and Author response

Decision letter https://doi.org/10.7554/eLife.85003.sa1
Author response https://doi.org/10.7554/eLife.85003.sa2

# Additional files

## Supplementary files

• Supplementary file 1. Supplementary phylogenetic and genomic engineering information.
(a) Sequence characteristics of ancestral nitrogenase subunits. (b) Host taxa of nitrogenase and outgroup dark-operative protochlorophyllide oxidoreductase homologs included for phylogenetic analysis. (c) Strains and plasmids used in this study. (d) Primers used in this study.

• MDAR checklist

## Data availability

Materials including bacterial strains and plasmids are available to the scientific community upon request. Phylogenetic data, including sequence alignments and phylogenetic trees, and the script for ancestral gene codon-optimization are publicly available at https://github.com/kacarlab/garcia_nif2023, (copy archived at swh:1:rev:c9b3cf5021e50b4a0995b3972ad81d5cedea4ed5). All other data are included as source data and supplementary files.

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
