## [Editor Report]

This manuscript reports valuable findings regarding the evolution of nitrogenases through ancestral sequence reconstruction and resurrection. The results are convincing and support the conclusions of the study, and highlight the historical constraints that have been acting on this enzyme. The findings will be of interest to people interested in enzyme evolution in general and particularly to those interested in the evolution of nitrogenases.

---

## [Decision Letter]

**Decision letter after peer review:**

Thank you for submitting your article "Nitrogenase resurrection and the evolution of a singular enzymatic mechanism" for consideration by *eLife*. Your article has been reviewed by 3 peer reviewers, and the evaluation has been overseen by a Reviewing Editor and Christian Landry as the Senior Editor. The following individuals involved in the review of your submission have agreed to reveal their identity: Matilda Newton (Reviewer #2); Christian B Macdonald (Reviewer #3).

Essential revisions:

– The reviewers raised questions about the impact of horizontal gene transfers on phylogenetic reconstructions. Further discussions led to the conclusion that this is probably not a major issue but it would be important to address this point in the paper.

– Reviewer 1 raised the issue that two distinct phylogenies were obtained with the same dataset as well as issues with the reconstruction methods implemented in the computational tools used. These would be important points to verify and clarify as needed.

*Reviewer #1 (Recommendations for the authors):*

I would strongly suggest the RaxML issue is explained in the methods. As a community, we should make sure that rigorous methods are used, especially since there is virtually no significant computational cost to the full algorithm anymore. As the Kacar lab is one of the leading groups in this field, it would be great to be clear about this.

If I misunderstood, and the authors' version of RaxML did in fact use the correct algorithm, this part of my public review should be deleted. It would still raise the question as to why the PAML and RaxML reconstructions differ, which would still need to be explained in the methods.

I did not understand exactly what the language model analysis shows. It is only mentioned in one sentence in the paper that does not explain the meaning of this analysis.

*Reviewer #2 (Recommendations for the authors):*

• Include a caveat that the reconstructed enzymes can only ever be hypotheses.

• I appreciated the insertion of the gene into the genome as opposed to plasmid-based expression

• ASR can only ever reconstruct the ancestor to extant enzymes, we can't rule out that there were other N2 fixation strategies competing pre-LUCA but they have left no trace.

• Why did you choose to use maximum likelihood inference instead of Bayesian?

• Figure 3A – please show the y-axis in the log.

• Could you please represent catalysis in units for kcat and KM? This would make the kinetics easier to compare to other enzymes, both extant and reconstructed.

• 4C please include WT.

• In light of the conclusion about a highly conserved mechanism, I would like to see more nuanced mechanistic studies. Pre-steady state kinetics; MD; pH studies.

• Please infer the age of the hypothetical ancestors expressing anc1 and anc2.

• Is there a notable difference between how the all-anc complexes (Anc1B) interact as opposed to the hybrids (Anc1A, Anc2)? Is there a notable difference in melting temperature or oligomeric state? This is discussed to a degree in the paragraph beginning line 323, but many of the statements are general and do not posit molecular explanations. What do the authors mean by "historical" amino acid substitutions?

• The discussion notes the surprising conservation of inhibition by H^+^ despite "substantial residue-level changes to the peripheral nitrogenase structure, as well as a handful within relatively conserved, active-site or protein-interface regions within the enzyme complex" Please elaborate on this, with specific attention to the active site. Are the residues involved/mechanisms known? Specify the mutations, how chemically conserved are they?

• I would be interested to see a discussion of potential ancestral promoters and expression levels. Expression levels are mentioned briefly in the results. I know this experiment must be compatible with the biological system used for the experiment (i.e. it would be impractical to also reconstruct ancestral promoters), but do the authors speculate that an ancestral nitrogenase would have been overexpressed to compensate for lower efficiency or that N2 fixation would merely have been rate-limited?

• For the uninitiated, please clearly introduce the evolution of metal ion dependence in nitrogenases.

• How do your results compare to enzyme reconstructions of a similar "age"?

• I find these results unsurprising. There is sufficient ASR literature to predict that a reconstructed enzyme will have comparable activity to the one or two extant enzymes it is compared to. When using a clade of conserved enzymes using a conserved mechanism, it is not surprising the ancestor conforms. What have we learned? It would be more interesting to probe these reconstructions for promiscuous activities or additional inhibitors. Are they easier to evolve than extant enzymes? If you reconstruct the whole pathway, does it behave differently? Does it act inefficiently and leak metabolites? Are other methods of fixation conceivable?

*Reviewer #3 (Recommendations for the authors):*

I have several questions and suggestions for the phylogenetic analysis that I do not believe will alter any of the results but may help with presentation or for a better understanding of the uncertainty with them.

• Why were these particular nodes picked for reconstruction? Are they the highest confidence reconstructions?

• Why was LG+G+F used? Was any model testing done?

• What tool was used to align the sequences?

• Why was ASR performed with RAxML and PAML both?

• What is the difference between the forak023 and forak013-14 files in the github repository? The topology of tree_forak013-14_branchsupport.tre seems to be the one in the manuscript, but there are no branch support values in that file.

As mentioned in the public comments, I worry whether HGT may cause issues during tree inference. I believe the simplest way to find out would be to reconstruct a gene tree for each individual nif gene and see how the trees differ. It could also be worth examining whether these or the tree in the manuscript agree with bacterial phylogenies.

The randomly-selected codon (de)optimization process is a nice inclusion. I was a little unsure about how it was performed, though – were codons swapped randomly until some metric was reached, or a fixed number, or some other procedure? Given the other controls, I do not expect this to change any results, but it would be a nice method for other groups to potentially use.

Given that the WT and ancestral sequences are 83% identical or higher (roughly akin to humans and mice), is the result that function is conserved surprising? Is it possible that these results say more about sequence (and functional) conservation, rather than a constraint? The UMAP embedding is an interesting approach, but makes this point in a different way, as the ancestral and WT sequences are extremely close in UMAP space. I believe some discussion of sequence.

---

## [Author Response]

Essential revisions:– The reviewers raised questions about the impact of horizontal gene transfers on phylogenetic reconstructions. Further discussions led to the conclusion that this is probably not a major issue but it would be important to address this point in the paper.– Reviewer 1 raised the issue that two distinct phylogenies were obtained with the same dataset as well as issues with the reconstruction methods implemented in the computational tools used. These would be important points to verify and clarify as needed.

The essential revision recommendations relate to (1) the impact of horizontal gene transfer on our reconstructions, and (2) our use of two distinct phylogenies.

Regarding horizontal gene transfer, we have addressed the bulk of these concerns in our response to Reviewer #3. Briefly, we do acknowledge that horizontal gene transfer has significantly shaped the evolution of nitrogenases, and we would not expect agreement between protein and species trees. However, these do not impact our central conclusions since we do not make inferences of species-level divergence. A potential impact is if the individual proteins in our concatenated alignments follow different evolutionary trajectories. For the reasons described in more detail in the following responses, we do not expect this to contribute significantly to the uncertainty associated with our specific reconstructions.

We thank Reviewer #1 for outlining the issue with RAxML, which we became aware of in the process of carrying out this study. We therefore repeated our study with PAML, which, as Reviewer #1 points out, appears to yield comparable experimental results. We have followed the reviewer’s recommendation to highlight the discrepancies in ancestral sequence reconstruction algorithms, so the broader community does not inadvertently use the incorrect algorithm.

Reviewer #1 (Recommendations for the authors):I would strongly suggest the RaxML issue is explained in the methods. As a community, we should make sure that rigorous methods are used, especially since there is virtually no significant computational cost to the full algorithm anymore. As the Kacar lab is one of the leading groups in this field, it would be great to be clear about this.

We fully welcome the reviewer’s recommendation to explain the RAxML v.8 issue in our Materials and methods. We initially used RAxML v.8 for ancestral sequence reconstruction (ASR) (e.g., following Aadland and Kolaczkowski, Genome Biol Evol, 2020). Despite RAxML v.8 documentation describing the algorithm as marginal reconstruction (as the reviewer notes), we were subsequently made aware that it is not the correct implementation. We therefore repeated our analysis using RAxML v.8 for tree reconstruction and PAML for ASR. As the reviewer also points out, ancestral sequences reconstructed from equivalent nodes in the RAxML and PAML analyses were very similar (~95% identical) and both exhibit the core N_2_ reduction mechanism described in the main text. Therefore, uncertainty associated with our use of the incorrect ASR algorithm does not impact the central findings of our study.

In our revised text, we clarify that RAxML v.8 does not implement full marginal ancestral sequence reconstruction and justify our repeated ASR analysis in more detail.

If I misunderstood, and the authors' version of RaxML did in fact use the correct algorithm, this part of my public review should be deleted. It would still raise the question as to why the PAML and RaxML reconstructions differ, which would still need to be explained in the methods.

The reviewer is correct that the RAxML version used did not implement the full algorithm, as outlined above.

I did not understand exactly what the language model analysis shows. It is only mentioned in one sentence in the paper that does not explain the meaning of this analysis.

Our aim with the language model analysis was not to test a specific hypothesis, but to visualize the protein sequence space occupied both by extant and ancestral nitrogenases. On one hand, it places the studied ancestors in the context of this available diversity, and also charts a “roadmap” for future studies that will navigate a broader swath of this sequence space. We have now clarified the text accordingly.

Reviewer #2 (Recommendations for the authors):• Include a caveat that the reconstructed enzymes can only ever be hypotheses.

We have included a new paragraph in our Discussion that addresses uncertainty in ancestral sequence reconstruction, including the fact that reconstructed ancestral enzymes represent hypotheses regarding the true ancestral state.

• I appreciated the insertion of the gene into the genome as opposed to plasmid-based expression

We’re glad the reviewer appreciated this crucial feature of our study.

• ASR can only ever reconstruct the ancestor to extant enzymes, we can't rule out that there were other N2 fixation strategies competing pre-LUCA but they have left no trace.

We agree that ASR is fundamentally limited to reconstructing the ancestors of whatever descendent enzymes have survived to the present. It is true that we cannot exclude the possibility that other, early-evolved nitrogen-fixing enzymes followed different strategies that were subsequently outcompeted. We address these possibilities in additional text in our Discussion, and have clarified text that discusses evolutionary constraints on nitrogen fixation strategies so as to capture these possibilities.

• Why did you choose to use maximum likelihood inference instead of Bayesian?

Both maximum-likelihood (ML) and Bayesian phylogenetic methods have previously been applied to nitrogenase evolutionary studies, for example in our (authors Garcia, Kaçar) earlier work (Garcia et al., Geobiology, 2020; Garcia et al., Genome Biol Evol, 2021) and others (Boyd et al., Front Microbiol, 2011). With either method, we observe that the general topology of nitrogenase trees is maintained. Additionally, previous work has demonstrated that Bayesian methods don’t necessarily generate more accurate ancestral sequences than ML methods (Hanson-Smith, Mol Biol Evol, 2010). Finally, given the greater computational expense of Bayesian methods – potentially weeks of computation using our resources with an alignment containing several hundred concatenated sequences – we elected to prioritize broader sequence sampling and used ML.

• Figure 3A – please show the y-axis in the log.

The plot for Figure 3A has been edited to show the y-axis in log.

• Could you please represent catalysis in units for kcat and KM? This would make the kinetics easier to compare to other enzymes, both extant and reconstructed.

We have used “specific activity,” (units of nmol product/nmol protein/s), which is kcat. Nitrogenase is a complex system for which Km is not an appropriate metric. A brief description is in the text and further details can be found in Harris et al., Biochemistry, 2019; 2022.

• 4C please include WT.

We have included data for WT in a revised Figure 4C.

• In light of the conclusion about a highly conserved mechanism, I would like to see more nuanced mechanistic studies. Pre-steady state kinetics; MD; pH studies.

Our central finding concerns a major aspect of nitrogenase mechanism (i.e., the reductive elimination/oxidative addition, “re/oa”, model for N2 binding and reduction). These conclusions are not dependent on a more exhaustive investigation of other properties relating to nitrogenase mechanism. We direct the reviewer to Harris et al., Biochemistry, 2019; 2022 which provide a more in-depth description of the re/oa model and kinetics, which is also cited in the main text.

• Please infer the age of the hypothetical ancestors expressing anc1 and anc2.

We did not perform time calibrations for our nitrogenase phylogeny, and age estimates based on species divergence are challenged by horizontal gene transfer (e.g., Parsons et al., Geobiology, 2021). Previous studies have estimated the timing of nitrogenase emergence (Parsons et al., Geobiology, 2021, Boyd et al., Geobiology, 2011), but more detailed constraints for the timeline targeted here are not presently available. We look forward to performing our own analysis in a future study.

• Is there a notable difference between how the all-anc complexes (Anc1B) interact as opposed to the hybrids (Anc1A, Anc2)? Is there a notable difference in melting temperature or oligomeric state? This is discussed to a degree in the paragraph beginning line 323, but many of the statements are general and do not posit molecular explanations. What do the authors mean by "historical" amino acid substitutions?

Our ability to copurify NifDK complexes from all ancestors (Anc1A, Anc1B, Anc2) and their exhibited activity in vitro suggests that there is no substantial distinction in their oligomeric states (i.e., they all form NifDK heterotetramers that interact with the NifH homodimer during the catalytic cycle). We did not investigate their melting temperatures.

Nevertheless, we do highlight the organism-level phenotypic differences observed between Anc1A and Anc1B strains, which likely stem from the additional substitutions (relative to WT) in the NifH and NifK proteins of Anc1B, which are not present in Anc1A. We interpret that these phenotypic outcomes might result from perturbed interactions within the complex (though evidently not enough to change the core oligomeric state), which can be more deeply explored in future work. We have now clarified these inferences in the Discussion.

By historical substitutions, we refer to differences between WT and ancestral amino acids at a given site. Upon reviewer’s feedback, we think that “ancestral substitution” might better capture this intended meaning. We have replaced this terminology and defined it at its first occurrence in the article.

• The discussion notes the surprising conservation of inhibition by H^+^ despite "substantial residue-level changes to the peripheral nitrogenase structure, as well as a handful within relatively conserved, active-site or protein-interface regions within the enzyme complex" Please elaborate on this, with specific attention to the active site. Are the residues involved/mechanisms known? Specify the mutations, how chemically conserved are they?

Our original Results text included a description of the sequence level differences observed between WT and Anc1/Anc2 ancestors. There, we highlight specific residues in functionally significant regions (e.g., I355V in the active site, within a loop considered important for cofactor insertion; F429Y and R108K within the NifD:NifK interface). A global visualization of sequence level differences relative to their conservation is shown in SI Appendix, Figure S3, and a listing of substitutions at relatively conserved positions is found in SI Appendix, Table S1. To our knowledge, the functional significance of many of these amino acid sites are not well characterized.

• I would be interested to see a discussion of potential ancestral promoters and expression levels. Expression levels are mentioned briefly in the results. I know this experiment must be compatible with the biological system used for the experiment (i.e. it would be impractical to also reconstruct ancestral promoters), but do the authors speculate that an ancestral nitrogenase would have been overexpressed to compensate for lower efficiency or that N2 fixation would merely have been rate-limited?

We agree that ancestral protein expression is worthy of study, though, to date, not well explored. As the reviewer suggests, we (authors Garcia, Kaçar) recently reported that expression levels of an ancestral and less active RuBisCO enzyme are increased relative to the WT enzyme (Kedzior et al., Cell Reports, 2022). However, we don’t see strong evidence of nitrogenase overexpression in the present study (Figure 3D).

Since expression levels in our experiments are dictated by regulatory mechanisms possessed by our model, *Azotobacter*, it’s challenging to infer whether our specific results would represent ancestral protein expression levels in an ancient host organism. We can speculate that expression of an ancestral, Mo-dependent nitrogenase might itself be ultimately limited by Mo availability, particularly early in Earth history when bulk marine Mo concentrations were extremely low (e.g., Anbar, Science, 2008). We envision that these possibilities can be tested in future genome engineering studies building off our presented experimental system.

We have acknowledged the role of protein expression in shaping ancestral phenotypes in additional text within the Discussion.

• For the uninitiated, please clearly introduce the evolution of metal ion dependence in nitrogenases.

We assume the reviewer is referring to a statement mentioning evolution of metal dependence in nitrogenases in our Introduction and agree that diversity of nitrogenase metal dependence is not well introduced. We now include additional text earlier in the introduction that describes the variability of nitrogenase metal dependence (i.e., relying on Mo, V, and Fe).

• How do your results compare to enzyme reconstructions of a similar "age"?

To our knowledge, our study represents the first laboratory reconstruction of nitrogenase enzymes. Therefore, we cannot yet compare our nitrogenase reconstructions to others of similar age.

• I find these results unsurprising. There is sufficient ASR literature to predict that a reconstructed enzyme will have comparable activity to the one or two extant enzymes it is compared to. When using a clade of conserved enzymes using a conserved mechanism, it is not surprising the ancestor conforms. What have we learned? It would be more interesting to probe these reconstructions for promiscuous activities or additional inhibitors. Are they easier to evolve than extant enzymes? If you reconstruct the whole pathway, does it behave differently? Does it act inefficiently and leak metabolites? Are other methods of fixation conceivable?

(Some of this text was also provided in response to similar comments from other reviewers).

We appreciate the reviewers’ comment and have revised relevant sections in our manuscript to clarify the novelty our study and include additional nuance into our discussions of conservation and constraints.

Nitrogenases are deep time enzymes and are a challenging target for engineering and functional study due to the number of protein components involved, their interactions with a broader cellular network, and their oxygen sensitivity (we now elaborate on these points in our Introduction). To date, only three, modern nitrogenase enzymes have recently been characterized with respect to their specific mechanism for N_2_ binding and reduction (the “reversible reductive elimination/oxidative addition” mechanism described in our article) (Harris et al., Biochemistry, 2019). Our study is therefore the first demonstration of this mechanism in nitrogenase ancestors and effectively doubles the number of nitrogenases that have been characterized to this degree.

Our broader point centers on the implications of this conservation in the evolution of biological nitrogen fixation strategies. Only one family of nitrogenase enzymes has evolved and survived to the present day. The comparable mechanistic features of extant nitrogenases and, now, ancestral nitrogenases, suggests that this one family has not only catalyzed N_2_ reduction for billions of years, but has achieved this incredibly challenging reaction in the same, specific manner.

The ecological importance of biological nitrogen fixation is on par with carbon fixation, though there are at least seven known pathways for achieving the latter. How life had become constrained to this particular N2 reduction mechanism remains an open question (particularly given several strategies for abiotic nitrogen fixation e.g., Cherkasov et al., Chem Engineer Process 2015; Dorr, Angew Chem Int Ed, 2003; Yung and McElroy, Science, 1979), but is one that can be further explored with the experimental approach presented here.

We agree that other aspects of nitrogenase functionality, including promiscuous activities, evolvability, and its specific interactions with other proteins involved in the nitrogen fixation pathway, are all excellent research targets that can also leverage our paleomolecular approach.

We have expanded our Discussion to include these key points.

Reviewer #3 (Recommendations for the authors):I have several questions and suggestions for the phylogenetic analysis that I do not believe will alter any of the results but may help with presentation or for a better understanding of the uncertainty with them.• Why were these particular nodes picked for reconstruction? Are they the highest confidence reconstructions?

The selected nodes are indeed well-supported (SH-like aLRT values ≈ 98-100). We have included this information in the Materials and methods. In addition, we chose a relatively conservative percentage identify threshold based on our (Garcia, Kaçar) prior resurrection studies to ensure we could recover active nitrogenase ancestors in our experimental system. We look forward to publishing the results of our ongoing work on older ancestral nodes across the nitrogenase tree.

We have updated the Results and Materials and methods with additional node selection rationale.

• Why was LG+G+F used? Was any model testing done?

Model testing performed by ModelFinder in IQ-TREE, now specified in Materials and methods.

• What tool was used to align the sequences?

MAFFT v7.450, now specified in Materials and methods.

• Why was ASR performed with RAxML and PAML both?

As we described in a response to Reviewer 1, we initially performed ASR with RAxML v.8, and constructed strains Anc1A and Anc2. However, due to concerns that this version of RAxML does not implement full marginal ancestral reconstruction, we repeated the analysis with PAML, constructing strains Anc1B (equivalent node to Anc1A). As we describe in our main text, Anc1A and Anc1B have the same set of descendant proteins and their NifD amino acid sequences are 95% identical.

• What is the difference between the forak023 and forak013-14 files in the github repository? The topology of tree_forak013-14_branchsupport.tre seems to be the one in the manuscript, but there are no branch support values in that file.

We thank the reviewer for noting this issue. We have updated the Github repository files and provided filenames that are more easily identifiable.

As mentioned in the public comments, I worry whether HGT may cause issues during tree inference. I believe the simplest way to find out would be to reconstruct a gene tree for each individual nif gene and see how the trees differ. It could also be worth examining whether these or the tree in the manuscript agree with bacterial phylogenies.

Indeed, as the reviewer points out, nitrogenase evolution is known to have been affected by significant HGT (e.g., Raymond et al., MBE, 2004; Parsons et al., Geobiology, 2021). We would therefore not expect protein trees and bacterial/archaeal species trees to agree, given our dataset. Discordance between the protein and species trees shouldn’t impact our main conclusions since we are not drawing inferences concerning species evolution from our protein tree. Previous work has also demonstrated that the individual nitrogenase genes have followed similar evolutionary trajectories (Raymond et al., MBE, 2004; Garcia et al., Genome Biol Evol, 2022), with the exception of H-subunit genes from V-nitrogenases (VnfH), some of which have diverged recently from molybdenum-dependent genes (NifH). However, given that (1) we are targeting a specific Nif lineage, (2) only one of our reconstructions includes an ancestral NifH gene, and (3) we nevertheless observe phenotypic consistency across multiple reconstructions, we do not expect issues stemming from HGT to have significantly impacted the present results.

Nevertheless, we concede that HGT is an important aspect of nitrogenase evolution and should be thoughtfully considered in ours and others’ future work, particularly for reconstructions that extend deeper in the nitrogenase phylogeny.

The randomly-selected codon (de)optimization process is a nice inclusion. I was a little unsure about how it was performed, though – were codons swapped randomly until some metric was reached, or a fixed number, or some other procedure? Given the other controls, I do not expect this to change any results, but it would be a nice method for other groups to potentially use.

We thank the reviewer for their comment and agree that providing more details on our codon optimization process would be helpful for the community. We used a semi-randomized optimization process to maximize identify between WT and ancestral nucleotide sequences. Ancestral and WT *A. vinelandii* protein sequences were compared using the alignment output of ancestral sequence reconstruction (RAxML or PAML). For sites where the ancestral and WT residues were identical, the WT codon was assigned. At sites where the residues were different, the codon was assigned randomly, weighted by *A. vinelandii* codon frequencies (Codon Usage Database, https://www.kazusa.or.jp/codon/). We have elaborated on our strategy in the Materials and methods and have added the relevant script to the public Kaçar Lab GitHub repository.

Given that the WT and ancestral sequences are 83% identical or higher (roughly akin to humans and mice), is the result that function is conserved surprising? Is it possible that these results say more about sequence (and functional) conservation, rather than a constraint? The UMAP embedding is an interesting approach, but makes this point in a different way, as the ancestral and WT sequences are extremely close in UMAP space. I believe some discussion of sequence.

(Some of this text was also provided in response to similar comments from other reviewers. It also seems that the reviewer comment here was truncated at the end. We’ve done our best to respond to what we expect were the reviewer’s intended recommendations).

We appreciate the reviewers’ comment and have revised relevant sections in our manuscript to clarify the novelty our study and include additional nuance into our discussions of conservation and constraints.

As the reviewer notes in their public comments, nitrogenases are indeed a challenging target for engineering and functional study due to the number of protein components involved, their interactions with a broader cellular network, and their oxygen sensitivity (we now elaborate on these points in our Introduction). To date, only three, modern nitrogenase enzymes have recently been characterized with respect to their specific mechanism for N2 binding and reduction (the “reversible reductive elimination/oxidative addition” mechanism described in our article) (Harris et al., Biochemistry, 2019). Our study is therefore the first demonstration of this mechanism in nitrogenase ancestors and effectively doubles the number of nitrogenases that have been characterized to this degree.

We agree that mechanistic conservation is likely an outcome of underlying sequence conservation. Our broader point centers on the implications of this conservation in the evolution of biological nitrogen fixation strategies. Only one family of nitrogenase enzymes has evolved and survived to the present day. The comparable mechanistic features of extant nitrogenases and, now, ancestral nitrogenases, suggests that this one family has not only catalyzed N_2_ reduction for billions of years, but has achieved this incredibly challenging reaction in the same, specific manner.

The ecological importance of biological nitrogen fixation is on par with carbon fixation, though there are at least seven known pathways for achieving the latter. How life had become constrained to this particular N_2_ reduction mechanism remains an open question (particularly given several strategies for abiotic nitrogen fixation e.g., Cherkasov et al., Chem Engineer Process 2015; Dorr, Angew Chem Int Ed, 2003; Yung and McElroy, Science, 1979), but is one that can be further explored with the experimental approach presented here.

We have expanded our Discussion to include these key points.